# Obesity-Associated Metabolomic and Functional Reprogramming in Neutrophils from Horses with Asthma

**DOI:** 10.3390/ani15131992

**Published:** 2025-07-07

**Authors:** Alejandro Albornoz, Beatriz Morales, Valentina Bernal Fernandez, Claudio Henriquez, John Quiroga, Pablo Alarcón, Gabriel Moran, Rafael A. Burgos

**Affiliations:** 1Institute of Pharmacology and Morphophysiology, Faculty of Veterinary Sciences, Universidad Austral de Chile, Valdivia 5090000, Chile; alejandroalbornoz.p@gmail.com (A.A.); beatriz.morales01@alumnos.uach.cl (B.M.); claudio.henriquez@uach.cl (C.H.); john.quiroga@uach.cl (J.Q.); pabloalarcon.u@gmail.com (P.A.); 2Graduate School, Faculty of Veterinary Sciences, Universidad Austral de Chile, Valdivia 5090000, Chile; vale.bernalfernandez@gmail.com

**Keywords:** equine asthma, obesity, neutrophils, metabolomics

## Abstract

Equine asthma is a chronic lung disease that causes breathing problems and involves inflammation driven by neutrophils. Obesity is becoming more common in horses and may worsen inflammatory conditions. This study compared obese and non-obese horses with asthma to understand how obesity affects neutrophil function. Obese horses showed higher levels of the inflammatory marker IL-1β and increased neutrophil activity. Their neutrophils also produced more reactive oxygen species and had an elevated expression of inflammatory genes. Metabolomic analysis revealed changes in how these immune cells process fats and amino acids, with increased levels of key molecules like itaconate and citraconic acid. These results suggest that obesity alters neutrophil metabolism, making them more reactive and possibly contributing to more severe airway inflammation. Managing obesity may therefore be important in treating equine asthma, and targeting neutrophil metabolism could offer new therapeutic opportunities.

## 1. Introduction

Equine asthma is a human asthma-like condition that develops in mature horses following stabling and exposure to dusty hay and straw [1]. The term “equine asthma” encompasses the previous names of recurrent airway obstruction (RAO) and inflammatory airway disease (IAD), which are now considered moderate-to-severe and mild forms, respectively, of a single disease; moreover, several authors suggest that equine asthma is the study model of neutrophilic asthma in humans [2,3,4]. *Aspergillus fumigatus*, an opportunistic fungus, is commonly found in environments surrounding horses and is considered to be one of the inciting agents in equine asthma [5]. Asthma-affected horses respond to this exposure by developing an increase in airflow resistance due to neutrophilic inflammation, mucus accumulation, and airway hyper-responsiveness, with a decrease in pulmonary function [1]. In general, airway inflammation in asthma horses involves the activation of pathogenic specific inflammatory cells, modulation of gene transcription factors, and release of inflammatory mediators [6].

Obesity is defined by the World Health Organization (WHO) as the abnormal or excessive accumulation of fat that can be harmful to the body, and is currently considered a global epidemic [7,8]. The worrying increase in overweight and obesity in today’s society is not limited to humans, with significant increases reported in companion and domestic animals [9,10]. In horses, numerous quantitative studies have also described a sustained growth in the prevalence of overweight and obesity [11,12,13], principally attributable to poor management practices, such as inappropriate feeding regimes and limited exercise [14]. Furthermore, obesity is often poorly recognised by owners, who tend to underestimate the body condition score (BCS) of their animals [15].

The relationship between obesity and human severe asthma appears to be weight-dependent, causal, partly genetic, and probably bidirectional [16]. Several studies suggest the existence of an excess risk of developing asthma in obese subjects compared with lean subjects, regardless of gender or age [17,18,19,20,21,22,23,24,25] and appears stronger for those with central adiposity [26,27]. Adipose tissue plays a role in the regulation of inflammation and immune function through the secretion of adipokines such as leptin (pro-inflammatory) and adiponectin (anti-inflammatory) [28,29]. In line with this, in asthmatic-obese subjects there is a rise in the serum concentrations of the pro-inflammatory adipokine leptin whereas adiponectin levels are decreased [30]. Furthermore, severe asthma is extremely common in patients with obesity [31]. A study reported that well over 50% of severe asthmatics were obese [32], suggesting a relationship between both diseases [33]. Furthermore, human obese patients tend to have more severe asthma that does not respond as well to conventional therapy compared with lean asthmatics [34]. Obese asthma patients are more at risk than lean asthma patients, as obesity has profound effects on the immune system and physiological function [35]. In horses, one study shows that obesity, defined by having a BCS of 7 points or higher, is a risk factor for the development of equine asthma [36].

The activation of immune cells exerts a profound influence on cellular metabolic pathways, enhancing the generation of energy and biosynthetic precursors required to sustain inflammatory processes and cell proliferation. During the last few years, much evidence has been generated about the role of cellular metabolism in the regulation of certain effector processes [37]. Studies mainly performed on macrophages and dendritic cells show how cells adapt their metabolism depending on certain signals sensed from their environment. Upon activation, these cells exhibit enhanced aerobic glycolysis (the Warburg effect), accompanied by a reduction in oxidative phosphorylation (OXPHOS), fatty acid oxidation (FAO), and Krebs cycle activity [38,39].

In the case of neutrophils, primary cytotoxic cells responding to inflammatory signals, in contrast to what was previously believed, do not represent only a homogeneous population of terminally differentiated cells but show high heterogeneity and phenotypic plasticity [40,41]. During maturation, neutrophils are primarily dependent on OXPHOS, FAO, and the pentose phosphate pathway (PPP). Subsequently, in a steady state, they rely mainly on glucose and lipid metabolism to sustain their survival and maintain redox balance. Upon activation in response to inflammatory stimuli or infections, neutrophils depend almost exclusively on aerobic glycolysis [42,43]. On the other hand, it has been shown that mitochondrial metabolism plays a key role in effector function by modulating chemotaxis, cytokine production, NET formation, and ROS generation [44,45,46,47].

Given the central role of neutrophils in equine asthma and their potential association with obesity in the development of more severe disease phenotypes, the aim of this study was to investigate whether differences in inflammatory markers exist and to evaluate their relationship with metabolic status and neutrophils functional activity in non-obese versus obese asthmatic horses in remission.

## 2. Materials and Methods

### 2.1. Animals

Three obese and three non-obese horses with naturally occurring asthma in clinical remission were included in this study. The diagnosis of severe asthma in the animals was established through a general clinical examination and bronchoalveolar lavage (BAL). At the time of diagnosis, all horses exhibited a clinical score above 15, based on a scoring system described by Lavoie at al. [48]. Cytological analysis of the BAL fluid confirmed a neutrophilic asthma phenotype in all cases, characterised by a neutrophil proportion exceeding 20%. During clinical remission, the horses showed no respiratory clinical signs, such as coughing, dyspnoea, or nasal discharge, and BAL cytology revealed a neutrophil percentage below 8%. The animals were aged between 5 and 12 years, belonged to the Chilean Criollo crossbreed, and comprised four mares and two geldings. None of the horses had received corticosteroid or bronchodilator treatment for at least two months prior to this study. All horses belong to the teaching and research herd of the Universidad Austral de Chile. Inclusion criteria for obese asthmatic horses include BCS score ≥7/9, based on a 9-point scale (wrinkles under the loin, difficulty in palpating the ribs, very soft fat around the root of the tail, fat around the withers and behind the shoulder, noticeable thickening of the neck, and fat deposited along the inner thigh) [49]. The animals were kept in the stables of the Veterinary Teaching Hospital or in their paddocks and were dewormed and clinically evaluated on a regular basis. All experimental procedures were approved by the Bioethics Committee for the Use of Animals in Biomedical Research of the Universidad Austral de Chile (n°497/2023).

### 2.2. Polymorphonuclear Isolation

The isolation of blood neutrophils was performed as previously described by our group [50]. Briefly, 10 mL of blood obtained by jugular venipuncture was placed in sterile tubes containing 1 mL of 3.8% *w*/*v* sodium citrate. Blood was placed on a discontinuous density gradient (Percoll, GE Healthcare, Uppsala, Sweden), with 4 mL of 85% Percoll in the bottom of a 15 mL tube and 4 mL of 70% Percoll above. After centrifugation (45 min, 670× *g*), the lower layer containing neutrophils, was aspirated for further processing. Granulocyte cell purity and viability were assessed by flow cytometry (BD FACS Canto II). Viability was assessed using an Annexin V assay as previously described by our group [50]. Cells were subsequently prepared for bioassays.

### 2.3. Sample Preparation for Metabolomics Analysis

Samples were prepared using methodology described in previous papers by this group [51,52]. Polymorphonuclear neutrophils (1 × 10^7^) were cryofractured with liquid nitrogen, and metabolites were extracted using cold acetonitrile/isopropanol/water (3:3:2, *v*/*v*/*v*) containing ribitol as an internal standard. After vortexing and centrifugation, the supernatant was dried in a SpeedVac (Thermo Fisher Scientific Inc, Waltham, MA, USA), reconstituted in acetonitrile/water (1:1), and centrifuged again. A second drying step was followed by addition of FAMEs (as retention index markers) and methoxyamine hydrochloride for oximation. Samples were incubated at 30 °C, then derivatised with MSTFA + 1% TMCS at 37 °C. The final derivatised extracts were transferred to glass vials for GC-MS analysis.

### 2.4. Metabolomics Analysis by GC-MS

The metabolomic analysis was previously published by our group [51,52]. Derivatised samples were analysed using an Agilent 7890B GC coupled to a 5977A mass selective detector in electron impact mode (Santa Clara, CA, USA), with a DB-5 column and splitless injection of 1 μL. The oven temperature was ramped from 60 to 325 °C over 37.5 min. Mass spectra were acquired in full-scan mode (m/z 50–600). Raw MS data were converted to ABF format and processed using MSDIAL 2.83 for peak detection, deconvolution, and alignment. Metabolite identification was based on spectral matching against NIST and Fiehn libraries using retention index (RI) and EI-MS similarity, applying thresholds of RI tolerance ≤ 2000, EI similarity ≥ 70%, m/z tolerance ≤ 0.5 Da, and RT tolerance ≤ 0.5 min.

### 2.5. Respiratory Burst Production

Reactive oxygen species (ROS) production by polymorphonuclear neutrophils (PMNs) was assessed using a luminol-dependent chemiluminescence assay adapted for equine neutrophils [53]. Briefly, neutrophils isolated from obese and non-obese asthmatic horses were resuspended at a concentration of 1 × 10^6^ cells/mL in Hank’s Balanced Salt Solution. The respiratory burst was triggered by stimulation with opsonised zymosan (0.1 mg/mL), followed by the addition of luminol (80 µM; Sigma Chemical Co., St. Louis, MO, USA) to each well. Negative controls consisted of unstimulated neutrophils incubated with luminol alone. Chemiluminescence, generated by the reaction between luminol and superoxide anion (O_2_^−^) or its dismutation products, was recorded over a 1 h incubation period at 37 °C using a Victor 2030 luminometer (Perkin Elmer, Waltham, MA, USA).

### 2.6. Enzyme Linked Immunosorbent Assay (ELISA) Analysis

Plasma concentrations of secreted IL-1β were quantified in obese and non-obese asthmatic horses using a commercial equine-specific ELISA kit (MyBioSource, San Diego, CA, USA), following the manufacturer’s instructions. The assay had a detection range of 31.25 to 2000 pg/mL and an inter-assay variability of 8%. Absorbance was measured at 450 nm using a Varioskan Flash multimode plate reader (ThermoFisher Scientific, Waltham, MA, USA). A four-parameter logistic (4PL) model was applied to generate the standard curve and to analyse the data, using GraphPad Prism software (version 9.1.0; GraphPad Software Inc., Boston, MA, USA).

### 2.7. Real Time Quantitative PCR Analysis

To evaluate the IL-1β gene expression in obese and non-obese asthmatic horses, neutrophils were plated onto 24 well plates (1 × 10^7^ cells/well), and then stimulated with 200 ng/mL of LPS for 2.5 h at 37 °C and 5% CO_2_, and finally incubated for 1 h with nigericin (5 µM). The RNA was extracted using Trizol reagent (Invitrogen, Carlsbad, CA, USA). The RNA (2 µg) was reverse transcribed using the superscript III enzyme, according to the manufacturer’s instructions (ThermoFisher Scientific). Gene amplification was performed with the RT-qPCR (StepOneTM, ThermoFisher Scientific, Waltham, MA, USA) method using SYBR green Master Mix (Takyon EU.UF RSMT B0701). The sequences of the primer pair used in this study are as follows: IL 1β (gene ID: 100034237): F-5′AGTACCCGACACCAGTGACA 3′; R- 5′GCCACAATGATTGACACGACA 3′ (product length 201 bp). The primers used in this study were designed by our laboratory using Primer Blast software https://www.ncbi.nlm.nih.gov/tools/primer-blast (accessed on 1 July 2023). Relative gene expression levels were normalised to the housekeeping gene *HPRT1* (gene ID: 100034149) (hypoxanthine-guanine phosphoribosyltransferase 1) using the following primers: forward 5′-GGTGAATACGGGACCTCTCG-3′ and reverse 5′-TGCATTGTTTTACCAGTGTCAA-3′ (product length 121 bp). Expression values were calculated using the comparative cycle threshold method (2^−∆∆Ct^ method) [19], with analysis performed in StepOne™ software version 2.3 (Applied Biosystems, Waltham, MA, USA).

### 2.8. Statistics Analysis

Graph generation and statistical analyses were performed using GraphPad Prism (GraphPad Software Inc., version 10.2.2, Boston, MA, USA) and SigmaPlot (Systat Software Inc., version 11.0, San Jose, CA, USA). The Kolmogorov–Smirnov test confirmed that the data were normally distributed. Comparisons between obese and non-obese asthmatic horses were made using the Student’s *t*-test, which enabled the assessment of differences in reactive oxygen species (ROS) production. Area under the curve (AUC) values for respiratory burst production, expression, and IL-1β levels were also calculated. A *p*-value of <0.05 was considered statistically significant. MetaboAnalyst software version 6.0 (http://www.metaboanalyst.ca (accessed on 11 May 2024)) was employed for all multivariate metabolomics analyses [54]. Metabolites that were frequently (>50%) below the limit of detection or had ≥50% missing values were excluded from the analysis. Ribitol was used as an internal standard for the normalisation of metabolite concentrations. Prior to statistical analysis, data were log-transformed and auto-scaled to approximate a Gaussian distribution [54]. Partial Least Squares-Discriminant Analysis (PLS-DA) was conducted using MetaboAnalyst v4.0.

## 3. Results

### 3.1. Asthmatic Obese Animals Exhibit Higher Levels of Circulating IL-1β

As shown in Figure 1, obese asthmatic horses exhibited significantly elevated systemic markers of inflammation compared to their non-obese counterparts. Basal serum concentrations of interleukin-1β (IL-1β) were markedly higher in the obese group (226.5 ± 21,979 pg/mL) than in the non-obese group (29 ± 218 pg/mL; *p* < 0.01) (Figure 1A), indicating a pronounced pro-inflammatory state. In addition, the total peripheral neutrophil count was significantly increased in obese asthmatic horses (5301 ± 1196 PMNs/mm^3^) compared to non-obese animals (3105 ± 360 PMNs/mm^3^; *p* < 0.05) (Figure 1B). Together, these findings suggest a potential association between obesity and enhanced innate immune responses in equine asthma, reflected by elevated circulating IL-1β levels and increased neutrophil counts. Nevertheless, these observations should be interpreted with caution, as they may not be generalisable and could be subject to individual and contextual variability.

### 3.2. Neutrophils from Obese Asthmatic Animals Display a Distinct Metabolic Phenotype

Untargeted metabolomic profiling identified a total of 139 metabolites. Upon classification into major chemical categories, these included 21 monosaccharides and oligosaccharides, 45 carboxylic acids and derivatives, 21 amino acids and their metabolites, 7 nucleotides and related compounds, 12 alcohols and polyols, and 33 metabolites falling under miscellaneous categories. Statistical interrogation of normalised data revealed a significant metabolic divergence between neutrophils isolated from obese and non-obese asthmatic animals. Supervised multivariate modelling via Partial Least Squares-Discriminant Analysis (PLS-DA) indicated that principal components 1 and 2 accounted for 18.7% and 51.3% of the total variance, respectively (Figure 2A). A hierarchical clustering heatmap constructed from the 55 most discriminant metabolites demonstrated clear segregation of the groups, underscoring the metabolic heterogeneity driven by the obese phenotype (Figure 2B). Figure 3 illustrates the relative abundance (log_10_ scale) of selected metabolites in leukocytes from non-obese and obese animals. A consistent increase in metabolite levels is observed across all panels in obese individuals compared to their non-obese counterparts. Notably, key intermediates of the tricarboxylic acid (TCA) cycle and immune regulation—such as citric acid, itaconate, and oxoproline—are significantly elevated, suggesting alterations in mitochondrial metabolism and antioxidant pathways. The observed increase in citraconic acid and itaconate may further reflect metabolic rewiring associated with immune cell activation.

Profound metabolic perturbations were particularly evident in neutrophils derived from obese asthmatic animals, with marked changes in pathways related to fatty acid metabolism—including oleic, palmitic, stearic, linoleic, arachidic, heptadecanoic, pentadecanoic, lignoceric, myristic, lauric, and capric acids. Amino acid metabolism was similarly affected, as reflected by increased levels of valine, isoleucine, glutamate, aspartate, histidine, lysine, tyrosine, asparagine, threonine, proline, glycine, phenylalanine, and serine (Figure 3). These changes may indicate enhanced protein turnover or shifts in nutrient sensing in immune cells. Taken together, the data suggest that obesity induces significant changes in metaboloma of neutrophils, characterised by an increased abundance of fatty acids, branched-chain amino acids, and TCA intermediates.

### 3.3. Metabolic Pathway Analysis

Pathway enrichment analysis of differentially expressed metabolites in neutrophils from obese versus non-obese asthmatic horses revealed significant alterations in multiple metabolic pathways. Using untargeted GC-MS metabolomics and KEGG-based pathway mapping with over-representation analysis and pathway topology scoring, several key pathways were identified as enriched (Figure 4). These included lipid metabolism (biosynthesis of unsaturated fatty acids, linoleic acid metabolism), amino acid biosynthesis (valine, leucine and isoleucine; phenylalanine, tyrosine, and tryptophan), and glutathione metabolism, indicating changes in membrane composition, immune-regulatory metabolites, and oxidative stress responses. Additionally, pathways involved in alanine, aspartate, and glutamate metabolism, as well as dicarboxylate metabolism, were enriched, suggesting shifts in nitrogen handling and mitochondrial function. These findings would suggest that obesity may influence neutrophil metabolic reprogramming in equine asthma, potentially enhancing inflammatory responses.

### 3.4. Neutrophils Isolated from Obese Asthmatic Animals Demonstrate a Heightened Responsiveness to Pro-Inflammatory Stimuli

Figure 5A illustrates the kinetics of the oxidative burst in peripheral blood neutrophils from non-obese and obese asthmatic horses following stimulation with opsonised zymosan. Time-resolved chemiluminescence profiles were used to monitor reactive oxygen species (ROS) production over a 10,000 s period. Neutrophils were either left unstimulated (C- or exposed to zymosan (C+), and the resulting chemiluminescence signals—expressed in arbitrary light units—reflect the dynamic ROS response. The traces represent the mean ROS production per condition for each group. The cumulative oxidative response, quantified as the area under the chemiluminescence curve (AUC), was significantly higher in neutrophils from obese horses compared to non-obese counterparts (*p* < 0.05) (Figure 5B). These results suggest an enhanced oxidative burst capacity in neutrophils from obese animals, supporting the hypothesis that obesity amplifies neutrophil reactivity in equine asthma and may thereby contribute to sustained airway inflammation and oxidative tissue damage.

Finally, Figure 6 shows the fold change in IL-1β mRNA expression in peripheral blood neutrophils isolated from non-obese and obese asthmatic horses following in vitro stimulation with lipopolysaccharide (LPS). Neutrophils from obese horses exhibited a significantly greater fold induction of IL-1β transcription compared to those from non-obese counterparts, reflecting an enhanced pro-inflammatory transcriptional response. These findings suggest that obesity may prime neutrophils for increased cytokine expression, which could contribute to the exacerbation of airway inflammation observed in equine asthma.

## 4. Discussion

The findings of this study would suggest a potentially relevant link between obesity, inflammation, and metabolic alterations in asthmatic horses. The significantly elevated serum IL-1β levels detected in obese asthmatic individuals suggest that this cytokine may contribute to disease exacerbation. This is consistent with evidence from human and animal studies demonstrating that altered visceral adipose tissue promotes systemic inflammation. Specifically, hypertrophic adipocytes are known to secrete pro-inflammatory cytokines such as TNF and IL-6, which drive the recruitment and activation of immune cells. Taken together, these observations support the hypothesis that obesity-related inflammation may play a mechanistic role in the progression of equine asthma [55,56].

IL-1β has been proposed as a key mediator in the crosstalk between macrophages and adipocytes during obesity [57,58]. Its neutralisation has been shown to attenuate obesity-induced inflammation, highlighting its role as an important regulator of inflammatory processes in this context [59]. Equine asthma serves as a valuable model for studying severe asthma in humans, with both conditions characterised by neutrophilic inflammation [1]. In humans, this inflammatory phenotype has been directly associated with obesity [31]. In the present study, we observed an increase in circulating neutrophil counts in the obese group of horses, indicating a potential link between higher BSC and systemic neutrophilia. This observation aligns with previous reports in humans [36,60,61]. Accordingly, elevated basal levels of IL-1β may contribute to the progression of equine asthma by amplifying airway inflammation through the promotion of neutrophil recruitment and activation.

Remarkably, neutrophils isolated from the peripheral blood of obese asthmatic horses exhibited clear metabolic alterations. The metabolomic analysis revealed substantial differences in the abundance of numerous metabolites when comparing neutrophils from obese versus non-obese asthmatic animals. These differences spanned several metabolic pathways, including linoleic acid metabolism; phenylalanine, tyrosine, and tryptophan biosynthesis; valine, leucine, and isoleucine biosynthesis; fatty acid biosynthesis; and glutathione metabolism. This distinct metabolic profile suggests that neutrophils in obese animals undergo metabolic reprogramming, likely in response to the obese state and/or chronic inflammatory stimuli. Such reprogramming has been associated with enhanced pro-inflammatory functions in neutrophils, including increased reactive oxygen species (ROS) production and the formation of neutrophil extracellular traps (NETs) [62]. These effector functions, while essential for host defence, may contribute to tissue damage and perpetuation of inflammation in chronic airway diseases. Therefore, the altered metabolic phenotype of neutrophils in obese asthmatic horses may play a critical role in amplifying and sustaining airway inflammation in this model of severe asthma.

Our study demonstrated elevated levels of branched-chain amino acids (BCAAs), specifically valine and isoleucine, as well as increased concentrations of positively charged amino acids such as histidine and lysine, and aromatic amino acids including tyrosine, phenylalanine, and tryptophan in neutrophils. This finding aligns with previous reports suggesting that neutrophils are a major source of the elevated plasma concentrations of BCAAs, aromatic, and positively charged free amino acids observed in diabetic patients [63,64,65,66]. The accumulation of BCAAs in plasma has been extensively documented and is strongly associated with insulin resistance in both obesity and type 2 diabetes [67,68,69,70,71,72,73,74]. BCAAs act as metabolic signaling molecules that influence glucose, lipid, and protein metabolism [75]. Moreover, BCAAs are potent activators of the mTOR signaling pathway [76] and play a crucial role in immune cell function by promoting lymphocyte proliferation and the activation of cytotoxic T cells [77]. The increased levels in these amino acids in neutrophils may reflect a shift towards alternative energy sources to meet heightened metabolic demands. This metabolic adaptation likely supports the functional activity of neutrophils, particularly under conditions of metabolic stress, as previously described [40].

Of particular interest in our study was the increased abundance of citrate and two of its downstream metabolites, citraconate and itaconate. Itaconate, in particular, is a metabolite known to play a key role in immune regulation and inflammation. Its accumulation has been well documented in macrophages exposed to lipopolysaccharide (LPS) or other inflammatory stimuli, as part of the metabolic reprogramming that occurs upon cellular activation [78,79,80]. Derived from cis-aconitate via the enzyme cis-aconitate decarboxylase (ACOD1), encoded by the Irg1 gene, itaconate synthesis reflects a shift towards an anaplerotic modification of the Krebs cycle. Itaconate exerts multiple immunomodulatory effects: it is a potent inhibitor of succinate dehydrogenase (SDH) [81], activates the anti-inflammatory transcription factors Nrf2 and ATF3 [82,83], and suppresses NLRP3 inflammasome activation [84]. Notably, infection models in mice have shown that neutrophils upregulate Irg1 more robustly than any other immune cell following exposure to *Staphylococcus aureus* or *Mycoplasma pneumoniae* [85,86]. In neutrophils, itaconate has been reported to impair glycolysis and ROS production, reduce bacterial killing capacity, and promote apoptosis. Conversely, itaconate-producing neutrophils have also been identified as regulators of both local and systemic inflammation in the context of trauma [87]. Therefore, the increased levels in itaconate observed in our study may reflect a cellular attempt to restrain inflammation. However, it remains to be clarified whether this compensatory response also contributes to the establishment of a more persistent pro-inflammatory phenotype in the setting of obesity-associated asthma.

The findings of our study reveal notable differences in the abundance of fatty acids—particularly saturated fatty acids—in obese animals. Obesity creates a nutrient-rich microenvironment, especially in terms of fatty acid availability [88] which may adversely affect immune cell function. Although mature neutrophils primarily rely on glycolysis to fulfil their effector roles, fatty acids are critical during their maturation and play a modulatory role in various neutrophil functions. For instance, oleic and linoleic acids—two unsaturated fatty acids found at elevated levels in our study—have been shown to induce NETs release in bovine neutrophils [89]. Similar dose-dependent effects have been reported with palmitic and oleic acids in human neutrophils [90], while a diet enriched in oleic and linoleic acids was shown to enhance neutrophil activity and modulate inflammation in rats [91]. Moreover, saturated fatty acids, including palmitic acid, have been demonstrated to activate neutrophils, with chain length appearing to influence this interaction [92]. In a recent study, mice fed a high-fat diet exhibited neutrophils with impaired phagocytic and bactericidal functions. Notably, neutrophils from obese animals displayed overexpression of genes involved in fatty acid metabolism—acyl-CoA thioesterase 1 (ACOT1) and carnitine palmitoyltransferase 1A (CPT1A)—as well as perilipin 2 (PLIN2), a marker of neutral lipid accumulation within lipid droplets, indicating enhanced fatty acid oxidation (FAO) and lipid storage [93]. In light of these findings, the diverse profile of fatty acid chain lengths observed in neutrophils from obese horses in our study may reflect both an increased uptake of circulating free fatty acids (FFA) and their active intracellular metabolism. This alteration in lipid handling may significantly affect neutrophil physiology and potentially exacerbate inflammatory responses in obese asthmatic horses. However, these effects are complex and warrant further investigation.

In accordance with their metabolic state, functional assays performed on neutrophils from obese asthmatic horses demonstrated a significant increase in their capacity to generate a respiratory burst upon stimulation with opsonised zymosan, as well as an enhanced inflammatory response following LPS treatment. In both instances, an augmented response to pro-inflammatory stimuli was observed. These findings are consistent with previous reports and suggest that neutrophils from obese individuals may exist in a pre-activated or “primed” state [94,95,96].

Equine asthma was studied during the remission phase, as this clinically stable state provides a particularly useful pathophysiological context to identify persistent immunological and metabolic alterations without the confounding effects of acute inflammatory responses characteristic of exacerbations. In this phase, subclinical changes in immune cell function—particularly neutrophils—can be detected, allowing exploration of processes associated with disease chronicity, such as residual inflammation or immunometabolic reprogramming. This approach is especially relevant for understanding the mechanisms that sustain the disease and for identifying early biomarkers of susceptibility to future exacerbations. Furthermore, remission offers more controlled experimental conditions, minimising variability introduced by intensive treatments or systemic stress, and thus, enables more consistent baseline measurements for comparative or longitudinal studies. Nevertheless, future investigations during clinical exacerbations will also be essential to deepen our understanding of the acute immunopathological and immunometabolic events, particularly those involving neutrophils.

In this study, equine asthma was investigated during the remission phase, as this clinically stable state provided a particularly valuable pathophysiological context for identifying persistent immunological and metabolic alterations, without the confounding effects of the acute inflammatory responses typically observed during exacerbations. During remission, we considered that subclinical changes in immune cell function—particularly neutrophils—could be detected, allowing the exploration of processes associated with disease chronicity, such as residual inflammation or immunometabolic reprogramming. This approach proved especially relevant for understanding the mechanisms that sustain the disease and for the potential identification of early biomarkers indicating susceptibility to future exacerbations. Furthermore, remission offers more controlled experimental conditions by minimising variability introduced by intensive treatments or systemic stress. As stated in the Materials and Methods section, the animals included in this study had not received corticosteroids or bronchodilators for at least two months prior to sampling, thereby allowing for more consistent baseline measurements for comparative purposes. Nevertheless, future investigations during clinical exacerbations will also be essential to deepen our understanding of acute immunopathological and immunometabolic events, particularly those involving neutrophils [1].

This study has several limitations that should be acknowledged. The small sample size *(n* = 6), although constrained by the availability of clinically well-characterised animals, limits the statistical power and generalisability of the findings. The cross-sectional design precludes causal inferences regarding the relationship between obesity, neutrophil metabolism, and asthma severity. Moreover, the analysis was confined to neutrophils, excluding the potential contribution of other immune cell populations. Functional validation of key metabolic findings—such as the role of itaconate in neutrophil activation—was not performed. Another limitation of this study is the absence of a non-asthmatic control group, which prevents us from determining whether the metabolic and inflammatory differences observed are specific to asthmatic animals or also occur in obese versus non-obese horses without asthma. Although this study focused on comparing obese and non-obese asthmatic animals, it is important to acknowledge that obesity alone could influence neutrophil function and metabolism, independently of the asthmatic condition. However, the chronic airway inflammation associated with asthma could induce immunometabolic responses that may be further exacerbated by obesity. Therefore, the differences described here—particularly the heightened inflammatory responsiveness and altered metabolic profiles in neutrophils—might be specific to the asthmatic context. Future studies including obese and lean non-asthmatic animals will help clarify whether obesity exerts similar immunometabolic effects in the absence of respiratory disease. This distinction is essential, as it would allow us to determine whether the obesity-associated neutrophil phenotype observed in this study acts as a disease-modifying factor in asthma or represents a general feature of obese horses. Finally, the lack of longitudinal data limits insights into dynamic changes or responses to therapeutic intervention. Future studies with larger cohorts and mechanistic approaches are warranted to corroborate and extend these observations.

## 5. Conclusions

The findings of this study demonstrate that obesity significantly influences the metabolic and inflammatory behaviour of neutrophils in horses affected by asthma. Obese horses exhibited elevated systemic IL-1β levels, increased circulating neutrophil counts, and heightened neutrophil responsiveness to pro-inflammatory stimuli, including enhanced oxidative burst and IL-1β gene expression. Metabolomic profiling revealed profound alterations in neutrophil metabolism, particularly in pathways related to fatty acid and amino acid metabolism, as well as the tricarboxylic acid cycle. Elevated concentrations of immunomodulatory metabolites such as itaconate and citraconic acid suggest that neutrophils in obese animals undergo metabolic reprogramming that may contribute to a sustained pro-inflammatory state. These results support the hypothesis that obesity promotes a distinct and potentially pathogenic neutrophil phenotype, which could exacerbate airway inflammation and impair disease resolution in equine asthma. Collectively, the data underscore the importance of addressing obesity as a key clinical and management factor in horses with asthma and highlight the potential of targeting immune cell metabolism as a novel therapeutic avenue in veterinary medicine.

## Figures and Tables

**Figure 1 animals-15-01992-f001:**
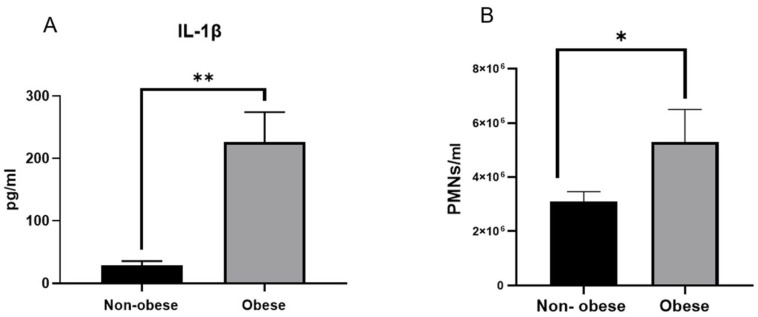
Circulating IL-1β levels and total neutrophil counts in non-obese and obese asthmatic horses. (**A**) Serum concentrations of interleukin-1β (IL-1β; pg/mL) were significantly higher in obese asthmatic horses compared to non-obese counterparts. (**B**) Total neutrophil counts (PMNs/mL) in peripheral blood were also significantly increased in obese horses. Bars represent mean ± SEM, *n* = 3. *p* < 0.05 (*), *p* < 0.01 (**).

**Figure 2 animals-15-01992-f002:**
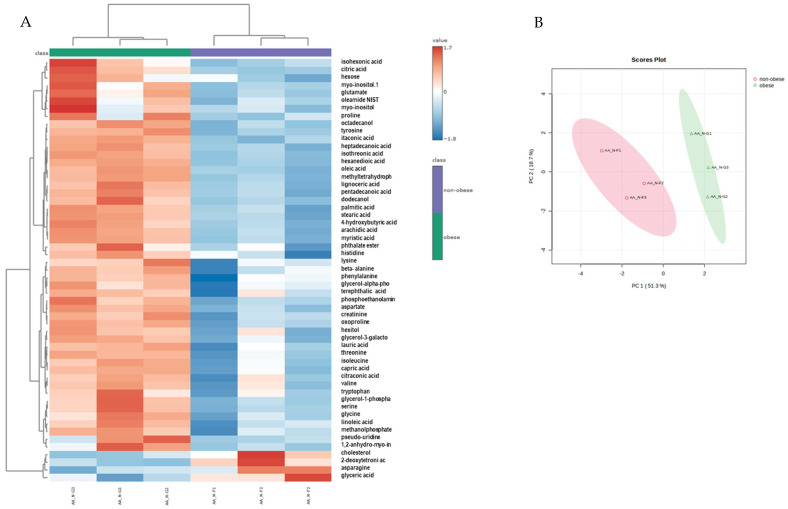
Heatmap and PLS-DA of neutrophil metabolomic profiles from non-obese and obese asthmatic horses. (**A**) Heatmap representing the relative abundance of the 55 metabolites with the lowest *p*-values, as determined by one-way ANOVA, in neutrophils isolated from non-obese and obese asthmatic horses. (**B**) Partial Least Squares-Discriminant Analysis (PLS-DA) score plot illustrating the separation of neutrophil metabolomic profiles between groups. Each point represents one biological replicate, *n* = 3.

**Figure 3 animals-15-01992-f003:**
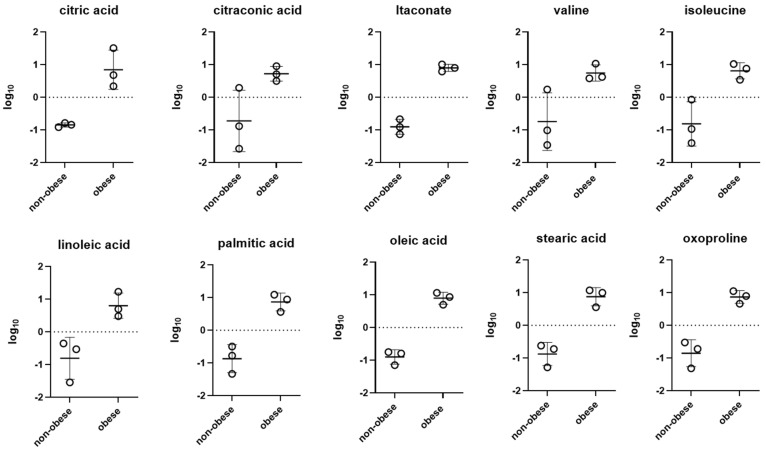
Log_10_-transformed relative abundance of selected metabolites in neutrophils from asthma-affected horses and obese asthma-affected horses. Obese animals consistently show higher levels of metabolites involved in TCA intermediates, amino acid, and lipid metabolism. Each dot represents an individual animal; horizontal lines indicate group medians, *n* = 3.

**Figure 4 animals-15-01992-f004:**
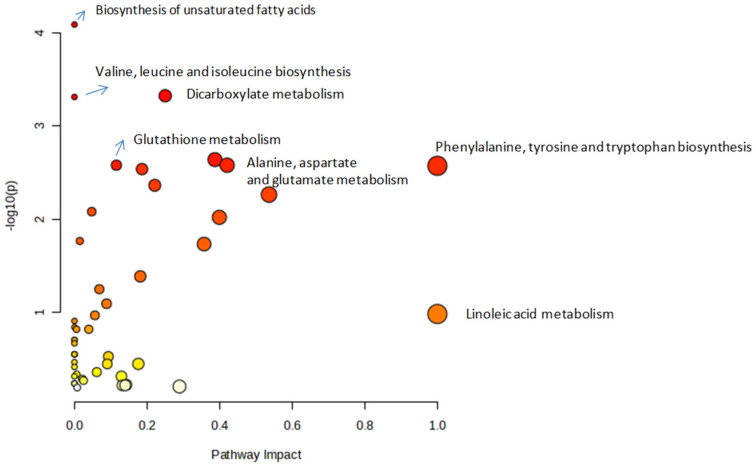
Metabolic pathways altered between neutrophils asthma-affected horses and obese asthma-affected horses. All matched pathways according to the *p*-values from enrichment analysis, and pathway impact values from the pathway topology analysis are shown, *n* = 3.

**Figure 5 animals-15-01992-f005:**
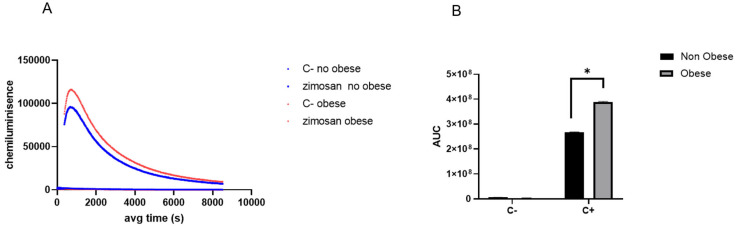
Temporal dynamics (**A**) and area under curve (AUC) quantification (**B**) of the oxidative burst in neutrophils from non-obese and obese asthmatic horses following zymosan stimulation. *n* = 3, * *p* < 0.05.

**Figure 6 animals-15-01992-f006:**
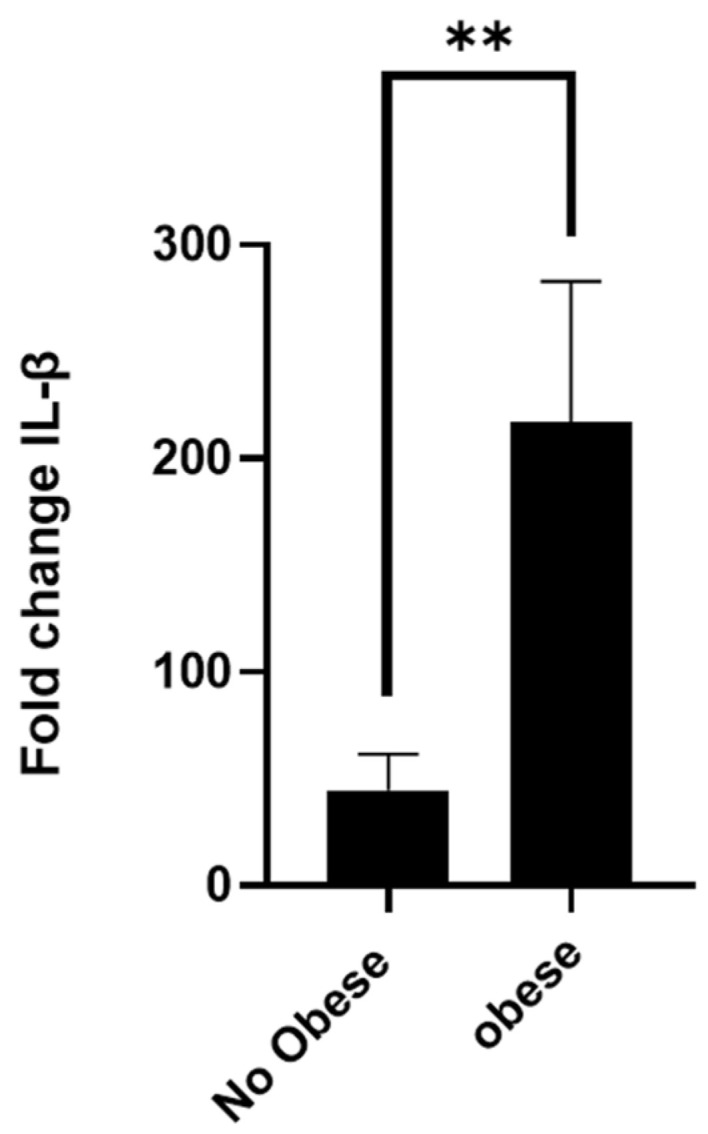
Relative IL-1β gene expression in neutrophils from non-obese and obese asthmatic horses following LPS stimulation. Gene expression was quantified by real-time PCR and normalised to a reference (housekeeping) gene. Data are presented as mean ± SEM. *n* = 3, ** *p* < 0.01.

## Data Availability

The data presented in this study are available on request from the corresponding author.

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
