# Peer review of "Obesity-Associated Metabolomic and Functional Reprogramming in Neutrophils from Horses with Asthma"

_animals, 2025, doi:10.3390/ani15131992_

Round 1

Reviewer 1 Report

Comments and Suggestions for Authors

This manuscript addresses an important and understudied intersection between obesity and respiratory inflammation in horses. The authors employed a comprehensive approach, combining in vivo measurements, functional neutrophil assays, and untargeted metabolomics, to check how excess adiposity alters neutrophil metabolism and inflammatory function. The findings have translational relevance to veterinary practice and comparative study. The principal limitation is the small sample size. Nonetheless, these data can be published and serve as valuable pilot results that can guide more expansive, definitive studies in the future.

The authors have thoughtfully acknowledged several important limitations of their work, including the small sample size. This is a promising study with novel insights into how obesity may reprogram neutrophil metabolism in equine asthma, however, addressing the following minor points will further strengthen the manuscript:

Primer characterization: Please provide full details for both IL-1β and HPRT qPCR assays, including primer sequences, target loci, expected amplicon lengths, annealing temperatures, and NCBI accession numbers for the reference sequences.

ELISA sensitivity: Report the lower limit of detection and dynamic range for the IL-1β ELISA kit used, as well as any intra- and inter-assay variability.

Remission versus exacerbation: All horses in this study were in clinical remission; it remains unclear whether the neutrophil metabolic phenotype you describe would persist or differ during active asthma exacerbations. A brief discussion of this point would help contextualize your findings. It may also be worth mentioning as a limitation of the study.

Author Response

Reviewer 1 :

This manuscript addresses an important and understudied intersection between obesity and respiratory inflammation in horses. The authors employed a comprehensive approach, combining in vivo measurements, functional neutrophil assays, and untargeted metabolomics, to check how excess adiposity alters neutrophil metabolism and inflammatory function. The findings have translational relevance to veterinary practice and comparative study. The principal limitation is the small sample size. Nonetheless, these data can be published and serve as valuable pilot results that can guide more expansive, definitive studies in the future. The authors have thoughtfully acknowledged several important limitations of their work, including the small sample size. This is a promising study with novel insights into how obesity may reprogram neutrophil metabolism in equine asthma, however, addressing the following minor points will further strengthen the manuscript:

  • Primer characterization: Please provide full details for both IL-1β and HPRT qPCR assays, including primer sequences, target loci, expected amplicon lengths, annealing temperatures, and NCBI accession numbers for the reference sequences.

R: The changes suggested by the reviewer have been implemented. “The sequences of the primer pair used in this study are as follows: IL 1β (gene ID: 100034237): F-5’AGTACCCGACACCAGTGACA 3’; R- 5’GCCACAATGATTGACACGACA 3’ (product length 201 bp). The primers used in this study were designed by our laboratory using Primer Blast software https://www.ncbi.nlm.nih.gov/ tools/prime blast/ (accessed on 1 July 2023). Relative gene expression levels were normalised to the housekeeping gene HPRT1 (gene ID: 100034149) (hypoxanthine-guanine phosphoribosyltransferase 1) using the following primers: for-ward 5′-GGTGAATACGGGACCTCTCG-3′ and reverse 5′-TGCATTGTTTTACCAG-TGTCAA-3′ (product length 121 bp). Line 197-207

  • ELISA sensitivity: Report the lower limit of detection and dynamic range for the IL-1β ELISA kit used, as well as any intra- and inter-assay variability.

R: The changes suggested by the reviewer have been implemented. “The assay had a detection range of 31.25 to 2000 pg/mL and an inter-assay variability of 8%”. Line 183-184

  • Remission versus exacerbation: All horses in this study were in clinical remission; it remains unclear whether the neutrophil metabolic phenotype you describe would persist or differ during active asthma exacerbations. A brief discussion of this point would help contextualize your findings. It may also be worth mentioning as a limitation of the study.

R: A paragraph has been added in the Discussion section explaining this point, line 477-493

Reviewer 2 Report

Comments and Suggestions for Authors

The manuscript "Obesity-Associated Metabolomic and Functional Reprogram- 2
ming in Neutrophils from Horses with Asthma" is very well written and easy to understand. The background of the study is well described and the focus of the study is very clear. Congratulations to the authors for this manuscript, nevertheless, information about the asthma diagnosis in the animals studied is missing completely. EA is a disease of middle aged to older horses, in this study the only information given is that they were older than 5 years. Please describe in detail how they were diagnosed with asthma, classify severity and subtype based on BAL cytology and discuss, why you studied animals in remission only. As the study includes only 6 horses, if the metabolic changes between remission and exacerbation and their possible relation to obesity were described and discussed. If this data on EA diagnostics is sound and included, I clearly recommend this manuscript for publication.

Author Response

Reviewer 2

The manuscript "Obesity-Associated Metabolomic and Functional Reprogram- 2 ming in Neutrophils from Horses with Asthma" is very well written and easy to understand. The background of the study is well described and the focus of the study is very clear. Congratulations to the authors for this manuscript, nevertheless, information about the asthma diagnosis in the animals studied is missing completely. EA is a disease of middle aged to older horses, in this study the only information given is that they were older than 5 years. Please describe in detail how they were diagnosed with asthma, classify severity and subtype based on BAL cytology and discuss, why you studied animals in remission only. As the study includes only 6 horses, if the metabolic changes between remission and exacerbation and their possible relation to obesity were described and discussed. If this data on EA diagnostics is sound and included, I clearly recommend this manuscript for publication.

R: The changes suggested by the reviewer have been implemented. “Three obese and three non-obese horses with naturally occurring asthma in clinical remission were included in the study. The diagnosis of severe asthma in these animals was established through a general clinical examination and bronchoalveolar lavage (BAL). At the time of diagnosis, all horses exhibited a clinical score above 15, based on a scoring system described by Lavoie at al. [48]. Cytological analysis of the BAL fluid con-firmed a neutrophilic asthma phenotype in all cases, characterised by a neutrophil pro-portion exceeding 20%. During clinical remission, the horses showed no respiratory clin-ical signs, such as coughing, dyspnoea or nasal discharge, and BAL cytology revealed a neutrophil percentage below 8%. The animals were aged between 5 and 12 years, be-longed to the Chilean Criollo crossbreed, and comprised four mares and two geldings. None of the horses received corticosteroid or bronchodilator treatment for at least two months prior to the study. All horses belong to the teaching and research herd of the Universidad Austral de Chile. Inclusion criteria for obese asthmatic horses include BCS score ≥7/9, based on a 9-point scale (wrinkles under the loin, difficulty in palpating the ribs, very soft fat around the root of the tail, fat around the withers and behind the shoul-der, noticeable thickening of the neck, and fat deposited along the inner thigh). Line 117-132

Reviewer 3 Report

Comments and Suggestions for Authors

Thank you for submitting this manuscript. It evaluates differences in metabolism of neutrophils between obese and non-obese asthmatic horses. While I agree with you that the small sample size can be a limitation, the information is still valuable as a starting point for future studies. I would suggest, though, giving more information about the study population: their age, breed and sex are important, in my opinion. Also, could you include more information about the clinical examination of the horses? You wrote they were in remission,  but for how long, and how was the asthma diagnosed, and how long before the inclusion?

A few small remarks:

Line 87-88: please delete "significantly associated", I would suggest leaving only "having a BCS of 7 points or higher was a risk factor for the development of equine asthma [36]"

Lines 89 and 105: please delete "On the other hand"

Lines 120-122: please add a closed parenthesis where appropriate

Line 207: please add "horses" after "asthmatic"

Author Response

Reviewer 3:

Thank you for submitting this manuscript. It evaluates differences in metabolism of neutrophils between obese and non-obese asthmatic horses. While I agree with you that the small sample size can be a limitation, the information is still valuable as a starting point for future studies. I would suggest, though, giving more information about the study population: their age, breed and sex are important, in my opinion. Also, could you include more information about the clinical examination of the horses? You wrote they were in remission,  but for how long, and how was the asthma diagnosed, and how long before the inclusion?

R: The changes suggested by the reviewer have been implemented. Line 117-132. Regarding the discussion on remission, a paragraph was added in lines 477-493

A few small remarks:

Line 87-88: please delete "significantly associated", I would suggest leaving only "having a BCS of 7 points or higher was a risk factor for the development of equine asthma [36]"

R: The changes suggested by the reviewer have been implemented

Lines 89 and 105: please delete "On the other hand"

R: The changes suggested by the reviewer have been implemented

Lines 120-122: please add a closed parenthesis where appropriate

R: The changes suggested by the reviewer have been implemented

Line 207: please add "horses" after "asthmatic"

R: : The changes suggested by the reviewer have been implemented

Round 2

Reviewer 2 Report

Comments and Suggestions for Authors

The authors have included a section about inclusion criteria and diagnostics performed to include asthma horses. They used well established methods and reference ranges, therefore I do not have any worries about the manuscript anymore and recommend it for publication. 

Author Response

Thank you for your positive assessment. We appreciate your careful review and are pleased to hear that the inclusion criteria and diagnostic methods addressed your concerns. We are grateful for your recommendation for publication.